# SP142 PD-L1 Assays in Multiple Samples from the Same Patients with Early or Advanced Triple-Negative Breast Cancer

**DOI:** 10.3390/cancers14133042

**Published:** 2022-06-21

**Authors:** Seung Ho Baek, Jee Hung Kim, Soong June Bae, Jung Hwan Ji, Yangkyu Lee, Joon Jeong, Yoon Jin Cha, Sung Gwe Ahn

**Affiliations:** 1Department of Surgery, Gangnam Severance Hospital, Yonsei University College of Medicine, Seoul 06273, Korea; holydante@yuhs.ac (S.H.B.); mission815815@yuhs.ac (S.J.B.); shevchencko@ish.ac.kr (J.H.J.); gsjjoon@yuhs.ac (J.J.); 2Institute of Breast Cancer Precision Medicine, Yonsei University College of Medicine, Seoul 06273, Korea; ok8504@yuhs.ac (J.H.K.); maytangerinetree@yuhs.ac (Y.L.); 3Division of Medical Oncology, Department of Internal Medicine, Yonsei University College of Medicine, Seoul 06273, Korea; 4Department of Pathology, Gangnam Severance Hospital, Yonsei University College of Medicine, Seoul 06273, Korea

**Keywords:** PD-L1, SP142, triple-negative breast cancer, atezolizumab

## Abstract

**Simple Summary:**

The IMpassion130 trial suggests that metastatic triple-negative breast cancer (TNBC) patients with PD-L1+ derived a clinical benefit from atezolizumab-combined treatment regardless of the sample collection time or origin. Therefore, if the PD-L1 test is positive at least once in multiple samples, the patient could have an opportunity to receive atezolizumab-based treatments. We aimed to know whether multiple PD-L1 testing might increase a rate of PD-L1+ in patients with TNBC. SP142 PD-L1 assays were performed in multiple samples from 77 patients in early TNBC. Multiple PD-L1 test using multiple samples raised the PD-L1+ rate more than a single biopsied sample test (68.8% vs. 37.6%, *p* = 0.00002). Among the group with metastatic TNBC treated with atezolizumab and nab-paclitaxel, PD-L1 assays were performed at least twice in 8/12 patients; 5/8 had heterogeneous results of PD-L1 assays. Consequently, a vigorous PD-L1 test using multiple samples was considered necessary in TNBC because a single test might be insufficient to represent the PD-L1 status.

**Abstract:**

Purpose: The discernible PD-L1 staining of tumor-infiltrating lymphocytes occupying ≥ 1% of the tumor area is considered SP142 PD-L1 positive for atezolizumab, and the PD-L1 status of multiple samples within a single patient could be discrepant. In this study, we evaluated the PD-L1 status by using the SP142 clone in serially collected matched samples from the same individuals with early or metastatic triple-negative breast cancer (TNBC). Method: the SP142 PD-L1 assay was performed using biopsies and surgical specimens from 77 patients with early TNBC. Among these patients, 47 underwent upfront surgery, and 30 underwent neoadjuvant chemotherapy (NAC) between biopsy and surgery. PD-L1 assays were performed at least twice in 8/12 (66.7%) patients with metastatic TNBC treated with atezolizumab and nab-paclitaxel. Results: Of the 47 patients who underwent upfront surgery, 15/47 (31.9%) had PD-L1+ on biopsied samples. PD-L1+ rates in the biopsy and surgical specimens increased to 66.0% (33 of 47) after subsequent surgery. Similarly, in the 30 patients with residual invasive cancer who underwent neoadjuvant chemotherapy, the PD-L1+ rate increased from 46.6% at baseline to 74.2% after surgery. In the 77 patients with early TNBC, multiple PD-L1 testing in the biopsies and surgical specimens significantly increased the number of patients with PD-L1+ compared with the number of patients with PD-L1+ assessed with initial biopsy samples alone (68.8% vs. 37.6%; *p* = 0.00002). Among the metastatic TNBC patients, those with constant PD-L1+ over 1% positivity in multiple samples showed a response which was longer than 12 months. Conclusions: Our findings reveal the heterogeneous SP142 PD-L1 expression in TNBC and suggest that PD-L1 evaluation in baseline biopsy might be insufficient to represent the PD-L1 status of whole tumors. In TNBC, vigorous PD-L1 examination using multiple available tumor samples could identify more patients eligible for immune checkpoint blockade.

## 1. Introduction

Programmed-death ligand-1 (PD-L1) status has been used to guide immune checkpoint blockades (ICBs), including atezolizumab and pembrolizumab, in metastatic triple-negative breast cancer (mTNBC) [1,2]. There are two PD-L1 assays approved as companion diagnostics to determine the use of ICBs in patients with mTNBC; the PD-L1 immunohistochemistry (IHC) 22C3 for pembrolizumab and the SP142 PD-L1 assay for atezolizumab. In the KEYNOTE-355 trial to test whether the addition of pembrolizumab to chemotherapy would improve the treatment outcome, the PD-L1 expression was evaluated by DAKO 22C3 and presented as a combined positive score (CPS). In the final analysis of this trial, pembrolizumab plus chemotherapy improved the overall survival vs. chemotherapy alone in mTNBC with a PD-L1 CPS of 10 or more [3]. Especially, 22C3 CPS enumerates the number of PD-L1+ cells including tumor cells, lymphocytes, and macrophages [2].

On the other hand, clinical trials with atezolizumab have adopted the SP142 PD-L1 assay to quantify PD-L1+ immune cells, and PD-L1 positivity was determined by the presence of the discernible PD-L1 staining of any intensity covering  ≥1% of tumor-infiltrating immune cells in the tumor area [1,4,5]. Intriguingly, the SP142 PD-L1 positive rates differed among the trials—41% in trials with mTNBC [1] and 46% and 56% in trials with non-mTNBC [4,5]—implying the heterogeneity of PD-L1 expression according to the sample type and disease status.

Importantly, phase III study of the IMpassion130 supports the benefit of atezolizumab in patients with SP142 PD-L1+ mTNBC [1,6]. In this study, the addition of atezolizumab to nab-paclitaxel informally displayed the overall survival (OS) benefits for patients with PD-L1+ mTNBC compared with nab-paclitaxel alone, although the OS benefit of atezolizumab-combined treatments was not statistically significant in the intention-to-treat population [6]. In addition, further analyses suggest that patients with PD-L1+ status derive clinical benefits from atezolizumab-combined treatments regardless of the sample collection time or sampling tissue site (primary vs. metastatic) [7]. This study included primary archival and fresh metastatic tissues to evaluate the PD-L1 expression in the context of a prospective analysis (62.6% with primary and 37.4% with metastatic tumors). Although the PD-L1+ rate was higher in primary tumor tissues than in metastatic samples (44.0% vs. 35.6%), improved treatment outcome by atezolizumab was observed irrespective of whether tissues were collected from primary or metastatic tumors. The results suggest that PD-L1+ status, regardless of the sampling site, could serve as in determining atezolizumab eligibility for the first line of treatment in patients with mTNBC. 

Thus, one positive result from the multiple PD-L1 testing of different samples could indicate the possibility of using atezolizumab-based treatments in patients with mTNBC.

Based on the literature, we hypothesized that multiple PD-L1 tests could increase the PD-L1 positivity rate in patients with TNBC. In this study, we performed an SP142 PD-L1 assay by using two or more samples from the same individuals with early or mTNBC. In early TNBC, samples were obtained from both biopsy and surgical specimens. In patients with mTNBC receiving atezolizumab and nab-paclitaxel as their first line of treatment, the PD-L1 status was evaluated in multiple serially collected samples, including archival primary or recent metastatic tissues.

## 2. Methods

### 2.1. Ethics Statement and Study Population

We obtained approval from the institutional review board which adheres to the Good Clinical Practice guidelines under the Declaration of Helsinki. Subsequently, through a retrospective search of our pathology database, we retrospectively identified all the patients at Gangnam Severance Hospital, Yonsei University College of Medicine, Seoul, Korea, between 1 January 2016 and 30 June 2021, with TNBC whose primary, recurrent, or metastatic breast cancer were subjected to the VENTANA SP142 PD-L1 assay. Seventy-seven patients whose PD-L1 status were evaluated more than twice in primary tumors with different sample types (biopsy and surgical specimens) were included. Of these patients, 47 underwent upfront surgery and 30 underwent neoadjuvant chemotherapy (NACT) between biopsy and surgery. Since June 2019, 12 patients with PD-L1+ mTNBC received atezolizumab and nab-paclitaxel as their first line of treatment. Figure 1 presents a summary of our patients.

Clinical and pathological data were obtained by reviewing electronic medical records. Patients diagnosed with recurrent or metachronous breast cancer were excluded. Stages were determined according to the 7th edition of the American Joint Committee on Cancer. The histological grading, PD-L1 immune cell score, and tumor-infiltrating lymphocyte (TIL) percentage score were evaluated in the biopsy and surgical specimens. TNBC was defined as the lack of estrogen receptor (ER), progesterone receptor (PR), and human epidermal growth factor receptor 2 (HER2) expression based on IHC staining [8]. For IHC examination, ER (clone 6F11; dilution 1:200; Leica Biosystems, Wetzlar, Germany), PR (clone 16; dilution 1:500; Leica Biosystems), and HER2 (clone 4B5; dilution 1:5; Basel, Switzerland) were used. In our study, ER and PR negativity were defined using a modified Allred score of ≤2 [9].

### 2.2. SP142 PD-L1 Assay and TIL Score

The PD-L1 and TIL percentage scores were assessed. We confirmed the PD-L1 expression status using the VENTANA PD-L1 SP142 assay (clone SP142; dilution prediluted; Ventana Medical System, Oro Valley, AZ, USA) PD-L1-specific antibody can be visualized using OptiView DAB IHC Detection Kit followed by the OptiView Amplification Kit according to the manufacturer’s protocols. Qualified benign human tonsil tissue was used as the control because it contains positive (moderate-to-strong PD-L1 staining noted in lymphocytes and macrophages in germinal centers, with diffuse staining in reticulated crypt epithelial cells) and negative (PD-L1 negative immune cells in the interfollicular regions with negative superficial squamous epithelium) staining elements for PD-L1 protein. Every staining run included tissue control for quality control. Tumor-infiltrating immune cells were scored as the proportion of the tumor area, including associated intratumoral and contiguous peritumoral stroma, occupied by the PD-L1 staining immune cells of any intensity. We assessed PD-L1 positivity in immune cells as a percentage and reported it in a binary manner with a 1% cutoff [1,10,11,12]. The tumor area occupied by ≥1% of immune cell positivity was considered PD-L1 positive.

Tumor infiltrating lymphocytes (TIL) levels were concurrently evaluated. We measured the TIL levels following the guidelines suggested by the International TIL Working Group [12]. Except polymorphonuclear leukocytes, other mononuclear cells including lymphocytes and plasma cells were counted. The TIL levels were reported as a percentage score for each case. For further analysis, 20% cutoff was applied to determine the low- and high-TIL groups. The PD-L1 SP142 assay and TIL counting were performed blindly, with no clinical information provided to the pathologist. The representative PD-L1 expression according to the TIL level was depicted in Figure 2. 

### 2.3. Atezolizumab and Nab-Paclitaxel Treatments

In 12 patients with mTNBC, atezolizumab and nab-paclitaxel were delivered as the first line of treatment. SP142 PD-L1 assays were conducted using available samples, regardless of the sampling site (primary or metastatic) or sample type (biopsy or excision). If any of the tests across multiple samples were positive, the patient was offered this regimen, as in the IMpassion-130 trial [1]. In a 28-day treatment cycle, patients received atezolizumab on days 1, 15 and nab-paclitaxel on days 1, 8, and 15. Atezolizumab is administered intravenously at the same dose of 840 mg but nab-paclitaxel is administered at a dose of 100 mg per square meter of body surface area. The administrations of these regimens were maintained until disease progression. Progression-free survival (PFS), defined as the interval between the first drug administration and disease progression or death, was evaluated.

### 2.4. Statistical Analyses

Differences in categorical variables between groups were calculated using Fisher’s exact test. A paired *t* test was used to compare the means of TIL levels and PD-L1 percentage scores between paired samples (biopsied and surgical tissues). Using univariable logistic regression analysis, we identified the factors affecting the change in PD-L1 status. The area under the curve (AUC) using receiver operating characteristic (ROC) curves was drawn with the Delong method [13] to test the ability of baseline TIL levels to predict PD-L1 change. The lm package was used for logistic regression analysis. All statistical analyses were performed using R software (https://www.r-projet.org, assessed on 16 June 2020; version 3.6.1). Statistical significance was set at *p* < 0.05.

## 3. Results

### 3.1. Study Population

We included 77 patients with non-mTNBC (47 with upfront surgery and 30 with NACT before surgery) who underwent PD-L1 tests in both the biopsied and surgical specimens and 12 patients with mTNBC treated with atezolizumab and nab-paclitaxel as their first line of treatment (Figure 1). The clinical and tumor characteristics of the 77 patients are shown in Appendix A. T1 (66.0%) and node-negative (87.2%) tumors were predominant in the upfront surgery group (Appendix A). By contrast, the majority of patients in the NACT group had T2-4 (90%) and lymph node metastasis (60.0%) at the initial diagnosis (Appendix A). The majority of these patients were treated with anthracycline and taxane sequential regimens (Appendix A).

### 3.2. Increased Rate of PD-L1 Positivity in Patients with Multiple PD-L1 Tests

In 77 patients with early TNBC, SP142 PD-L1 assays were performed twice for both the biopsy and surgical specimens. In both cohorts, the PD-L1 positivity rate tended to be higher in surgical specimens than in biopsied specimens: from 32% to 62% in the upfront surgery group and from 47% to 57% in the NACT group (Table 1). Among those, half of the PD-L1-negative results in the biopsied sample were PD-L1 positive results in the surgical specimen (Figure 3A,B). Altogether, half of the negative PD-L1 biopsied samples were converted to a positive PD-L1 status in the surgical samples (Figure 3C). 

Because the PD-L1 status within an individual can be determined as positive if any of the tests are positive, we compared the rates of PD-L1 positivity between single tests in biopsied samples and multiple tests in biopsied and surgical specimens. In both cohorts, the PD-L1-positive rate increased in the multiple tests compared with the single assays: it increased from 32% to 66% in the upfront surgery group (Figure 3A, *p* = 0.0018) and from 47% to 73% in the NACT group (Figure 3B, *p* = 0.0641). Consequently, in all 77 patients, multiple PD-L1 tests using both the biopsy and surgical specimens showed a significantly higher PD-L1+ rate than a single test using biopsied samples alone (69% vs. 38%; Figure 3C, *p* = 0.0002).

### 3.3. Factors Associated with the Conversion of PD-L1 Status from Negative to Positive According to Sample Type

In the upfront surgery group, half the PD-L1-negative tumors in the biopsied sample were PD-L1-positive at surgery without any treatments (Figure 2A). Thirteen of 15 PD-L1-positive tumors at biopsy were still PD-L1-positive tumors at surgery (Figure 2A). We investigated the factors affecting PD-L1 shifting (negative → positive) in the initial PD-L1-negative patients (*n* = 32) in the upfront surgery group because chemotherapy could alter the biology of the residual tumor and tumor microenvironment [14,15,16], affecting the PD-L1 status of surgical specimens after NACT.

Among the baseline characteristics, logistic regression analysis revealed that only the TIL score of biopsied samples was associated with the positive conversion of PD-L1 (odds ratio 1.11; 95% CI 1.04–1.27; Table 2). The optimized cutoff of TIL was identified as 20% to predict the PD-L1 positive shift using the ROC curve (Appendix A). The AUC for the TIL score was 0.779 (95% CI 0.637–0.921; Appendix A). The odds ratio of high TIL (≥20%) was 19.29 (95% CI 2.83–393.46) in predicting the positive conversion of PD-L1 (Table 2).

### 3.4. Changes in TIL and PD-L1 Scores between Paired Biopsied and Surgical Samples

Additionally, we compared the TIL and PD-L1 scores between the paired biopsy and surgical specimens (Figure 4). In the upfront surgery group, the TIL and PD-L1 percentage scores were significantly higher in the surgical specimens (Figure 4A; *p* = 0.0118 and *p* = 0.0310, respectively; paired *t* test). In the NACT group, the TIL score significantly decreased after chemotherapy in the surgical specimen (Figure 4B, top; *p* = 0.0001), whereas the PD-L1 score was numerically elevated without statistical significance (Figure 3 and Figure 4 down; *p* = 0.1042).

### 3.5. Multiple PD-L1 Tests in Patients Treated with Atezolizumab

We included 12 patients with mTNBC who received the first line of atezolizumab and nab-paclitaxel during the study period (Figure 5). The clinical and pathological characteristics are presented in Appendix A. Eleven patients had relapsed mTNBC except for one patient with de novo mTNBC. SP142 PD-L1 assays were performed at least twice using available archival tissues from primary or recurrent tumors in eight patients (66.7%). The results of the PD-L1 tests were heterogeneous in five patients (Figure 5, black arrow), and the median PFS was 3 (range, 1–8) months. In contrast, PFSs were longer than 12 months in the three patients who had consistently high (>1%) PD-L1 positivity across multiple samples (Figure 5, red arrowhead).

## 4. Discussion

We found that the PD-L1 positive rate could be elevated if the tests were performed in equal to or more than 2 samples and compared with the single test with biopsied samples alone in patients with early TNBC. Notably, specimens from upfront surgery showed a significant positive conversion in PD-L1 status. This finding suggests that multiple PD-L1 tests using available samples, including biopsied and resected specimens, could increase the PD-L1 positivity rate and provide additional eligibility for atezolizumab-combined treatment. Eight of the twelve mTNBC patients in this study had an SP142 PD-L1 test at least twice for the first line of atezolizumab-combined treatment.

Because the PD-L1 expression is spatially heterogeneous among various tumors [17], whether small tissue samples are sufficient to represent the PD-L1 status of the entire tumor has been controversial. Because of the spatial heterogeneity of PD-L1 expression, an assumption is that the PD-L1 status in biopsied samples might be underestimated compared to that in resected specimens. It has been reported that the discordance in PD-L1 status between biopsied and surgical specimens was notable: up to 48% in non-small cell lung cancer (NSCLC) [18,19]. Furthermore, a meta-analysis conducted in NSCLC showed that the positive rate of SP142 PD-L1 in biopsied samples was significantly lower than that in surgically resected samples [20]. Because the SP142 PD-L1 assay in NSCLC determines PD-L1 positivity by using both tumor cell and immune cell expression, a direct comparison with PD-L1 positivity in TNBC might be different. However, our study of early TNBC also showed higher SP142 PD-L1 positivity in the surgical specimen than in the biopsied sample (32% compared to 60%; Table 1). 

The spatial heterogeneity of PD-L1 expression could be associated with the TIL score, which reflects the volume of tumor-localized immune cells. As shown in Figure 3A, the TIL score in the resected specimen was significantly higher than that in the biopsied sample, indicating that the TIL score may be underestimated in small-biopsied samples. This finding is also concordant with our previous study, which showed that the TIL score was approximately 4.4% higher for resected specimens than for biopsied specimens, regardless of the molecular subtypes [11]. This finding could be applied to the PD-L1 percentage score because a positive correlation between TIL and PD-L1 expression has been observed [21,22]. Furthermore, we identified that high-TIL levels (≥20%) in biopsied samples were associated with the positive conversion of PD-L1 status, which suggests that re-testing the PD-L1 assay could result in PD-L1 positivity if a high-TIL-tumor was negative for PD-L1.

In the NACT cohort, a decrease in TIL score after NACT was observed (Figure 3B). In this cohort, the TIL scores were evaluated in residual invasive tumors after NACT, which confers chemotherapy resistance. This could be explained in the context of findings in the literature that TIL levels were lower in metastatic sites than in the primary site [23]. However, the PD-L1 positivity rate and percentage score tended to increase in residual tumors, which implied that immune cells in residual tumors were more likely to express PD-L1 (Table 1 and Figure 3B). Our findings provide clinical evidence for a change in immune dynamics after NACT and a rationale for several ongoing trials evaluating the addition of ICBs in residual TNBC [24,25].

Apart from widening the eligibility for ICBs using multiple PD-L1 testing, the clinical response to ICB-combined treatment is more important. In the IMpassion130 trial by Emens et al., the clinical benefit of atezolizumab-based treatments was consistent regardless of the tested sample site (either primary or metastatic) [26]. However, it remains unknown whether ICB-based therapy offers a comparable benefit in patients with varied PD-L1 results in recurrent and metastatic tumors. In this study, only patients with consistent PD-L1 positivity showed prolonged PFS. Further studies investigating the clinical outcomes of patients with a discordant PD-L1 status across multiple samples are warranted.

The limitation of this study is the small study population. The statistics had to be reflective of this, as we could not perform subgroup analyses in various ways as the sample size in a given category would have been way too small. Given that it is not an easy task to collect TNBC samples, we can draw meaningful findings in our early TNBC cohort despite the small sample size. On the contrary, a statistical analysis based on the results of multiple PD-L1 tests was restricted in the cohort with mTNBC as only 12 patients with mTNBC treated with atezolizumab were included. Therefore, the clinical benefit of atezolizumab treatment, even in patients with heterogeneous PD-L1 expression, should be verified in a larger cohort. In addition, we only used the SP142 clone to determine the PD-L1 status. Thus, it is unclear whether our findings could be applied to other PD-L1 assays, such as the DAKO 22C3-based combined positive score, which is a companion diagnostic for pembrolizumab in mTNBC. Studies have suggested that various PD-L1 assays [26,27] are not analytically interchangeable. Despite these limitations, considering a dismal prognosis of mTNBC, increasing the chance of ICB-based treatments through multiple PD-L1 tests in these patients is worthwhile.

## 5. Conclusions

In conclusion, we illustrated the heterogeneity of SP142 PD-L1 expression and the lack of representativeness of a single biopsy of tumors in addressing the PD-L1 status in TNBC. Vigorous PD-L1 examination using multiple available tumor samples should be encouraged to enable patients with TNBC to benefit from ICB-based treatment.

## Figures and Tables

**Figure 1 cancers-14-03042-f001:**
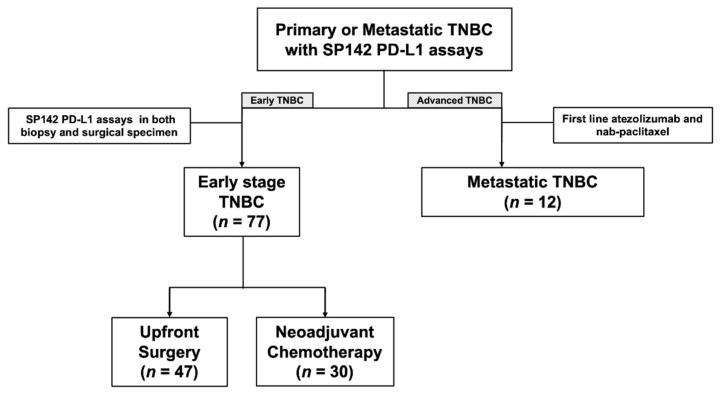
Diagram of the study population.

**Figure 2 cancers-14-03042-f002:**
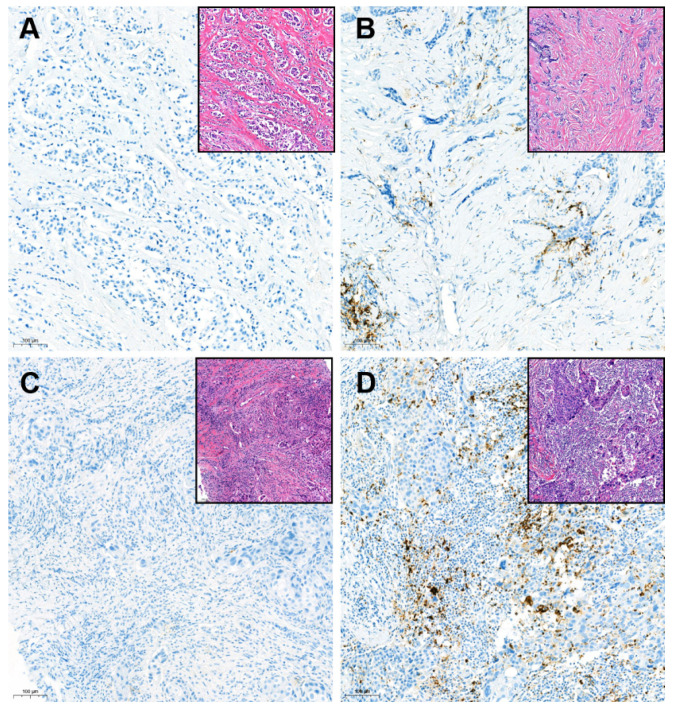
Representative cases for SP142 PD-L1 immunohistochemistry in tumors with variable tumor-infiltrating lymphocytes levels (×100 magnification): (**A**) Negative PD-L1 in the low-tumor-infiltrating lymphocyte (TIL) tumor; (**B**) Positive PD-L1 in the low-TIL tumor; (**C**) Negative PD-L1 in the high-TIL tumor; (**D**) Positive PD-L1 in the high-TIL tumor; Inlet pictures present the matched area of H&E images. The cutoff of TIL for the low- and high-TIL groups was defined as 20% in this study.

**Figure 3 cancers-14-03042-f003:**
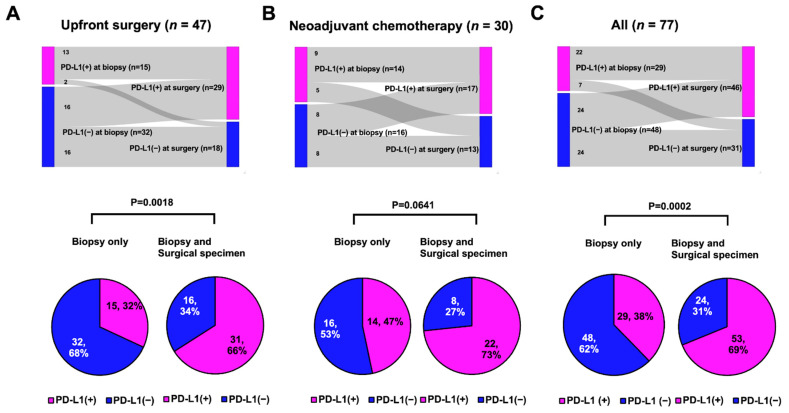
PD-L1 status between biopsied and surgical samples (tops) or rate of PD-L1 positive according to multiple samples (bottom): (**A**) Upfront surgery group: PD-L1-positive rates significantly increased from 32% to 66% (*p* = 0.0018, Student’s *t* test); (**B**) Neoadjuvant chemotherapy group: PD-L1-positive rates tended to be increased from 47% to 73% (*p* = 0.0641, Fisher’s exact test); and (**C**) All patients: PD-L1-positive rates were significantly increased from 38% to 69% (*p* = 0.0002, Fisher’s exact test).

**Figure 4 cancers-14-03042-f004:**
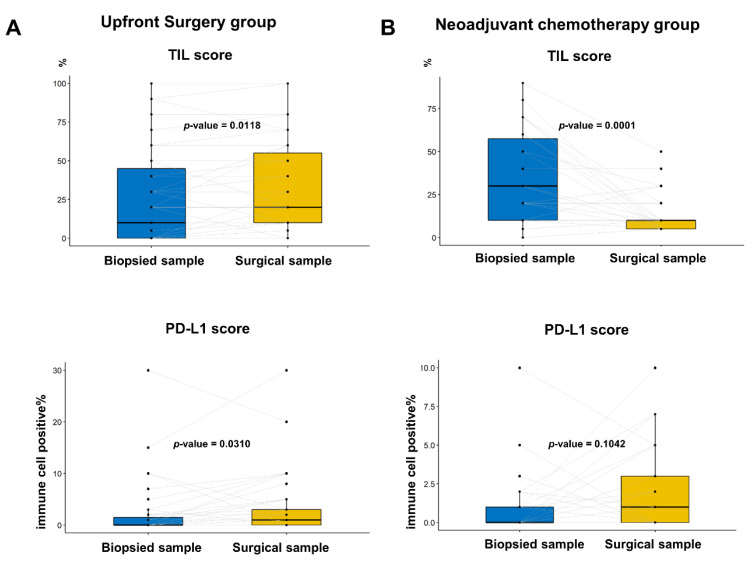
Comparison of percentage PD-L1 score and tumor-infiltrating lymphocyte (TIL) between the paired biopsied and surgical samples in the same patients: (**A**) In the group with upfront surgery, the TIL and PD-L1 score was significantly higher in the surgical samples than in biopsied samples (*p* = 0.0118 and *p* = 0.0310, respectively; paired *t* test); and (**B**) In the group with neoadjuvant chemotherapy, the TIL score was significantly lower in surgical samples than in biopsied samples, whereas the PD-L1 score tended to be elevated without statistical significance (*p* = 0.0118 and *p* = 0.310, respectively; paired *t* test).

**Figure 5 cancers-14-03042-f005:**
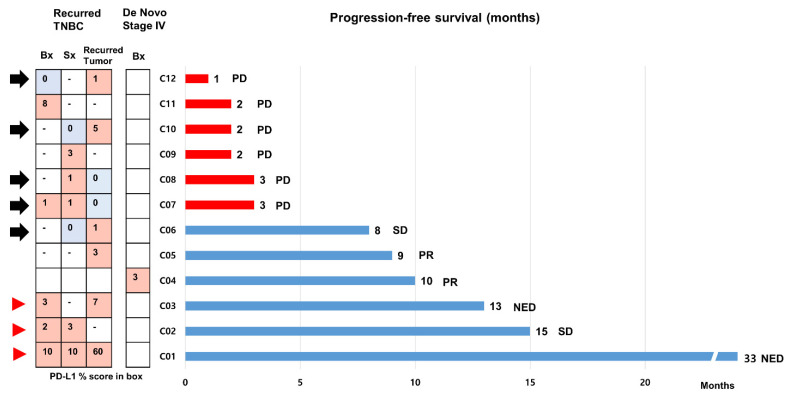
Progression-free survivals of 12 patients with metastatic triple-negative breast cancer treated with first line atezolizumab and nab-paclitaxel. We assessed the progression-free survival (PFSs) of 12 patients with metastatic triple-negative breast cancer (mTNBC) who received first line atezolizumab and nab-paclitaxel using a bar chart. Eleven patients had relapsed mTNBC, except for one patient with de novo mTNBC. Of these, SP142 PD-L1 assays were performed at least twice using archival tissues from primary or recurrent tumors in 7 patients. The black arrow indicates cases with heterogeneity in PD-L1 status across the samples. PFSs were longer than 12 months in 3 patients who had strong, constant PD-L1 positivity across multiple samples (red arrowhead). Abbreviations: NED, no evidence of disease; PD, progression of disease; PR, partial response; SD, stable disease.

**Table 1 cancers-14-03042-t001:** PD-L1 status and means of percentage score between multiple samples in each group.

	Upfront Surgery (*n* = 47)			Neoadjuvant Chemotherapy (*n* = 30)		
	Biopsied Sample	Surgical Sample	*p*	Biopsied Sample	Surgical Sample	*p*
PD-L1 status			0.007			0.606
Negative	32 (68%)	18 (40%)		16 (53%)	13 (43%)	
Positive	15 (32%)	29 (60%)		14 (47%)	17 (57%)	

Abbreviation: PD-L1, Programmed-death ligand-1.

**Table 2 cancers-14-03042-t002:** Logistic regression analysis for the positive conversion of PD-L1 in the surgical specimen (*n* = 32).

Variable	Odds Ratio	95% CI	*p*
Continuous TIL	1.11	1.04–1.27	0.029
TIL ≥ 20%	19.29	2.83–393.46	0.010
Histologic grade at biopsy	1.143	0.279–4.683	0.853
Clinical T stage	1.798	0.515–6.272	0.357
Clinical N stage	1.963	0.086–44.976	0.673

Abbreviations: PD-L1, programmed-death ligand-1; CI, confident interval; TIL, tumor-infiltrating lymphocytes.

## Data Availability

The data that support the findings of this study contain clinical outcomes for which institutional review board (IRB) approval is required before analysis. Therefore, these data are not publicly available. The data will be provided to authorized researchers who have obtained IRB approval from their institution and Gangnam Severance Hospital, Yonsei University, Seoul, Korea. For data access requests, please contact the corresponding authors, namely Dr. Y.J.C. or S.G.A., email address: yooncha@yuhs.ac or asg2004@yuhs.ac.

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
