# Peer review of "SP142 PD-L1 Assays in Multiple Samples from the Same Patients with Early or Advanced Triple-Negative Breast Cancer"

_cancers, 2022, doi:10.3390/cancers14133042_

Round 1

Reviewer 1 Report

The manuscript titled “SP142 PD-L1 assays in multiple samples from the same patients with early or advanced triple-negative breast cancer” describes that the authors would like to suggest the clinical practice should use multiple patients’ samples to diagnose the personal tumor condition to perform the specific treatment to benefit the patients, for example, to test SP142 PD-L1 expression. The followings are some concerns and comments have been pointed out that the authors may want to consider.

Major Concerns and Comments:

1.      Line 49 introduction section: This part should be enriched with more information.

2.      Line 102 section 2.2: Provide IHC images.

3.      Line 109: I don’t think it’s a good idea to cite two articles from “Your own” publications to support the statement “The presence of discernible PD-L1 staining of any intensity covering 1% of the tumor area was considered PD-L1 positivity [1,9-12].” I highly suggest just keeping one reference from the two of the following: either reference [10] or reference [11].

4.      Line 102 section 2.2: At least a brief description of the protocol should be provided.

5.      Lines 181-185: Provide shreds of evidence to the statement “We investigated factors affecting PD-L1 shifting (negative positive) in the initial PD-L1 negative patients (n=32) in the upfront surgery group because chemotherapy could alter the biology of the residual tumor and tumor microenvironment, affecting the PD-L1 status of surgical specimens after NACT.

6.      Line 172 Figure 2: a) There is no need to include so many color legends since they are under the same panel. I’d suggest the authors just keep one. b) Please make it clear how did you use Student’s t-test to analyze two pie graphs?  c) Please add color legends to the pink column and blue column.

7.      Line 190: Where is Figure S1?

Minor Concerns and Comments:

1.      Line 22: Please use italic p as it refers to a p-value. Check throughout the manuscript.

2.      Lines 23-24: Please rephrase “PD-L1 assays were performed at least twice in 8 of 12 and 5 have heterogeneity.”

3.      Line 35: I’d suggest the authors use “15/47” to make it clearer. Check throughout the manuscript.

4.      Line 41: The right half “)” is missing.

5.      Line 79 section 2. Methods: All the reagents used in this study should be listed with detailed resource information. For the antibodies, the dilution ratio should provide.

6.      Line 145 Table S1: Please consistent the format, in Table S1 is the upper case “N”, in Table S2 is the lower case “n”. Check throughout the manuscript.

7.      Line 169 Table 1: The meaning of the number in the brackets should be mentioned. Please adjust the table to make it looks better and update it to make it easier to read.

8.      Line 170: “PD-L1” has been used lots of times before line 170, please define it the first time where it appears.

9.      Line 205 Figure 3:  a) Please add sample numbers. b) Please homogenous the consistency of “p” or “p-value” throughout the manuscript. I’d suggest the authors just use the letter p that’s fine.

10.  Line 225: The “st” should be superscript for “1st”.

11.  Line 288: A space is needed between the words “the” and “SP142”.

Author Response

Reviewer #1 (Comments to the Author):

Major Concerns and Comments

  1. Line 49 introduction section: This part should be enriched with more information

Response) We enriched more information about 22C3 PD-L1 assay, which is approved as the companion diagnostics to determine the use of pembrolizumab in metastatic TNBC. We cited two key references with the KEYNOTE-355 trial, which showed overall survival benefit in the final analysis. We mentioned that 22C3 PD-L1 CPS counts the number of PD-L1+ cells including tumor cells, which is specifically different from the SP142 PD-L1 assay.

             Moreover, we added details of Impassion-130 trial in relation to our study. Percentages of collected primary and metastatic samples were added; also, concrete percentages for prevalence of PD-L1+ samples according to sampling sites for PD-L1 tests were displayed in the introduction.  

  1. Line 102 section 2.2: Provide IHC images

Response) We added images of representative cases for SP142 PD-L1 IHC in tumors with variable tumor-infiltrating lymphocytes levels: (A)(B) Negative or Positive PD-L1 in low TIL tumors (C)(D) Negative or Positive PD-L1 in high TIL tumors. Inlet pictures contain matched area of H&E images; The cutoff of TIL for low- and high TIL group was defined as 20% in this study.

  1. Line 109: I don’t think it’s a good idea to cite two articles from “Your own” publications to support the statement “The presence of discernible PD-L1 staining of any intensity covering ≥1% of the tumor area was considered PD-L1 positivity [1, 9-12].” I highly suggest just keeping one reference from the two of the following: either reference [10] or reference [11].

Response) Except reference 10, we removed other references.

  1. Line 102 section 2.2: At least a brief description of the protocol should be provided.

Response) We further described regarding the PD-L1 SP142 assay protocol including information of secondary antibody and amplification kit. As recommended, every staining run included benign tonsil tissue as a tissue control because tonsil contains positive and negative tissue elements for the PD-L1 proteins. Also, brief interpretation of PD-L1 SP142 assay for TNBC was also described in the same section.

  1. Lines 181-185: Provide shreds of evidence to the statement “We investigated factors affecting PD-L1 shifting (negative positive) in the initial PD-L1 negative patients (n=32) in the upfront surgery group because chemotherapy could alter the biology of the residual tumor and tumor microenvironment, affecting the PD-L1 status of surgical specimens after NACT.

Response) Several studies supports our statement. The MIMOSA-1 study concluded that neoadjuvant chemotherapy (NACT) induces significant immune microenvironment changes in TNBCs and showed that PD-L1 and TIM-3 expression post-NACT may provide prognostic information [1]. Other studies evaluated the impact of systemic therapy on tumor microenvironment and provide evidence that chemotherapy could modulate tumor immunogenicity and generally influence protagonist of an adaptive immunity [2,3]. Although the implicated mechanism remain unclear, some authors proposed a transcriptional up-regulation of PD-L1 gene by oncogenic signaling pathways such as mTOR/PI3K/AKT [4,5]. Accordingly, several studies exhibited that PD-L1 expression could be increased after chemotherapy in various malignancies [6-9].”

Among these references, we cited first 3 references at the end of the statement.  

  1. Line 172 Figure 2: a) There is no need to include so many color legends since they are under the same panel. I’d suggest the authors just keep one. b) Please make it clear how did you use Student’s t-test to analyze two pie graphs?  c) Please add color legends to the pink column and blue column.

Response) a) We unified the color legends as pink and blue. b) Sorry for this mistake. We corrected the legends and clarified that we used Fisher’s exact test. c) We added color legends to the each column. Overall, we recreated Figure 2 according to your critique (Now Figure 3).

  1. Line 190: Where is Figure S1?

Response) Thanks for checking this. We uploaded a supplementary file not attached to figure S1. We re-inserted Figure S1 in the supplementary file. Sorry for our mistake.

Minor Concerns and Comments

  1. Line 22: Please use italic p as it refers to a p-value. Check throughout the manuscript.

Response) We corrected those accordingly.

  1. Line 23-24: Please rephrase “PD-L1 assays were performed at least twice in 8 of 12 and 5 have heterogeneity”

Response) Thank you for your comments. We rephrased this statement as below:

PD-L1 assays were performed at least twice in 8/12 patients; 5/8 had heterogeneous results of PD-L1 assays.

  1. Line 35: I’d suggest the authors use “15/47” to make it clear. Check throughout the manuscript.

Response) We corrected those throughout the manuscript.

  1. Line 41: The right half “)” is missing

Response) We corrected it.

  1. Line 79 section 2. Methods: All the reagents used in this study should be listed with detailed resource information. For the antibodies, the dilution ratio should provide.

Response) All the information including clone, dilution ratio, and manufactures were further provided in the methods as follows.

ER (clone 6F11; dilution 1:200; Leica Biosystems, Wetzlar, Germany)

PR (clone 16; dilu-tion 1:500; Leica Biosystems)

HER2 (clone 4B5; dilution 1:5; Basel, Switzerland)

PD-L1 (clone SP142; dilution prediluted; Ventana Medical System, Oro Valley, AZ, USA)

  1. Line 145 Table S1: Please consistent the format, in Table S1 is the upper case “N”, in table S1 is the lower case “n”. Check throughout the manuscript.

Response) We use “n” uniformly throughout the manuscript.

  1. Line 169 Table 1: The meaning of the number in the brackets should be mentioned. Please adjust the table to make it looks better and update it to make it easier to read.

Response) The number in the brackets means percentage, and you can find it in revised manuscript. We also adjusted the position of the table to look better.

  1. Line 170: “PD-L1” has been used lots of times before line 170, please define it the first time where it appears.

Response) In the first line of the introduction section (Line 50), we defined programmed-death ligand-1 (PD-L1) at first appearance. 

  1. Line 205 Figure 3: a) Please add sample numbers. b) Please homogenous the consistency of “p” or “p-value” throughout the manuscript. I’d suggest the authors just use the letter p that’s fine

Response) We unified “p” throughout the manuscript.

  1. Line 225: The “st” should be superscript for “1st”

Response) Sorry for this mistake. We corrected it.

  1. Line 288: A space is needed between the words “the” and “SP142”

Response) Thanks for the detailed review. Because the contents of the section in the revised manuscript were modified, this simple mistake does not exist now.

We thank the reviewer for your careful reading of our manuscript and your helpful comments and suggestions, which were valuable and helpful for revising and improving our manuscript. We now hope that our revisions meet your requirements for publication.

Reviewer 2 Report

The authors provide evidence linking multiple PDL1 testing in multiple patients to determine the choice of therapy. They tested 77 patients multiple times and showed that repeat testing might increase the chance of PDL1 positivity, although deemed negative before. The proposed idea will help to decide on PDL1-based therapy in cancer, particularly in triple negative breast cancer.

Some comments are:

Was there any issue with testing itself? Did any of the same tissue test positive upon multiple testing? Did the authors observe any positive PDL1 staining in areas of the tumor that were previously negative? Is it possible to provide IHC pictures and show if similar areas for patients?

In their multiple testing, were there samples that tested negative after testing positive previously? How do the authors explain the higher positivity in surgical samples outside side of treatment? Could it be because of the larger tumor area? Is it possible to have multiple biopsies from various tumor areas to confirm positivity (as Figure 3 shows)? It would be essential to discern between PDL1 upregulation and technical issues in PDL1 testing. Was there any correlation with lymphocyte infiltration before and after surgery or treatment? In figure 2A, were these patients subjected to multiple testing?

Line 18 can be explained better or rewritten in the simple summary

The author may want to include some details about the IMpassion130 trial and its relation to their study.

The authors can cite the names of the packages used in R for logistic regression analysis.

Author Response

Jun 15, 2022

Reviewer #2 (Comments to the Author):

Was there any issue with testing itself? Did any of the same tissue test positive upon multiple testing? Did the authors observe any positive PDL1 staining in areas of the tumor that were previously negative? Is it possible to provide IHC pictures and show if similar areas for patients?

Response)  As mentioned in the method section, we did not perform multiple PD-L1 testing in the same tumor tissue. Since every staining run included tissue control and qualified the staining quality, multiple testing on the same tissue is unnecessary. Multiple PD-L1 testing was performed in the same patients of different sites (primary and metastasis) or of specimen from different procedure (biopsy and resection). For the early TNBC patients, multiple tests were performed on biopsy and matched surgically resected tissue, and there was a significant increase in the number of patients with PD-L1-positivity on combined results from biopsy and resected tissue, compared to PD-L1 positivity assessed in initial biopsy sample alone. (68.8% vs 37.6%, p=0.0002).

For the metastatic TNBC patients, multiple tests were performed in the primary breast tissue and tissue obtained from metastatic sites.

We inserted representative IHC pictures as figure 2 in section 2.2; it is newly created and displays 2x2 cases according to PD-L1 status and tumor-infiltrating lymphocyte counts.

In their multiple testing, were there samples that tested negative after testing positive previously?

Response) As shown in the revised Figure 3C, there were 48 patients who showed PD-L1 (-) in the initial biopsy specimen. Among those, 50.0% (24/48) were converted to a positive PD-L1 status in the surgical sample. Also, 29 patients showed PD-L1 (+) in the initial biopsy specimen. Among those, 24.1% (7/29) were converted to a negative PD-L1 status in the surgical sample.

How do the authors explain the higher positivity in surgical samples outside side of treatment? Could it be because of the larger tumor area?

Response) The technique of SP142 PD-L1 assay accounts the higher positivity of PD-L1 in surgical samples as it measures PD-L1 expression on TILs. In high TIL tumor, larger tumor area could provide more chance to increase PD-L1 positivity. However, PD-L1 expression has known to be heterogeneous, it cannot be simply explained only with extent of tumor area. As PD-L1 interpretation is performed in both intratumoral area and peritumoral stroma, surgical specimen is likely to have abundant peritumoral stroma compared to biopsy specimen. Taken together, it is expected that PD-L1 positivity would be expected in surgical specimens than in biopsied samples.

Is it possible to have multiple biopsies from various tumor areas to confirm positivity (as Figure 3 shows)? It would be essential to discern between PDL1 upregulation and technical issues in PDL1 testing.

Response) First, PDL1 testing is well-developed assays for companion diagnosis and under strict quality control during every staining procedure. In this study, we found that surgical samples were more likely to be positive for PD-L1 compared to the biopsy alone. This implies that single biopsy alone would be insufficient to represent the PD-L1 status of whole tumor. Rather than “PD-L1 upregulation”, it appears to be more appropriate to explain that multiple biopsy from variable tumor area would more likely to represent the whole tumor area and reflect the more accurate PD-L1 status of tumor, in line with same context above. (increased PD-L1 positivity in surgical specimen than in biopsy specimen)

Was there any correlation with lymphocyte infiltration before and after surgery or treatment?

 Response) Although not in the manuscript, when compared to a total of 77 early TNBC patients, the TIL count was statistically decreased in the surgical specimen compared to the biopsied sample (p=0.036; Paired T test). The decrease of TIL before and after surgery would depend on the effect of neoadjuvant chemotherapy group. As shown in figure 3 (now figure 4), the TIL count of the surgical specimen increased in the upfront surgery group compared to the biopsied sample, but in contrast, the TIL count of the surgical specimen in the neoadjuvant chemotherapy group decreased dramatically. The takeaway is that chemotherapy has a much more pronounced effect on TIL count than surgery itself. The answer for major comment 5 from the reviewer #1 also supports this finding.

In figure 2A, were these patients subjected to multiple testing?

Response) In all 77 early TNBC patients, their PD-L1 status was confirmed by performing PD-L1 test on biopsied samples and surgical specimens. That is, yes, these patients were subjected to multiple testing.

Line 18 can be explained better or rewritten in the simple summary

Response) Thank you for your comments. We could rewritten this line to “Therefore, if the PD-L1 test is positive at least once in multiple samples, the patient could have an opportunity to receive atezolizumab-based treatments.”

The author may want to include some details about the IMpassion130 trial and its relation to their study

Response) According to the patients inclusion criteria of the IMpassion 130 trial protocol, if multiple tumor samples (e.g. archival specimen and tissue from relapsed disease) can be obtained from a patient, the sample with the maximum PD-L1 score becomes the patient’s PD-L1 score. That means that even if PD-L1 is negative initially, it can be accepted to use atezolizumab if it is confirmed as positive in later test. If PD-L1+ is confirmed even once in multiple samples, it would be belonged in the PD-L1+ group in the IMpassion 130 trial.

Actually, in the Impassion 130 trial, the PD-L1+ rate was higher in primary tumor tissues than in metastatic samples (44.0% versus 35.6%); also, improved treatment outcome by atezolizumab was observed irrespective of whether tissues were collected from primary or metastatic tumors. We added this information in the introduction of the manuscript.

Our study showed that the PD-L1+ rate increases when the tests are performed with multiple samples. Thus, we could say that the vigorous PD-L1 test is important and will offer more chance for the use of ICBs.

The authors can cite the names of the packages used in R for logistic regression analysis.

Response) We used lm packages for logistic regression analysis. We added this statement in the statistics.

We thank the reviewer for your careful reading of our manuscript and your helpful comments and suggestions, which were valuable and helpful for revising and improving our manuscript. We now hope that our revisions meet your requirements for publication.

Sincerely,

Sung Gwe Ahn, MD, PhD

Reviewer 3 Report

This is an interesting original research paper. Methods, results and conclusion are well described.

As mentioned in the limitations part by the authors themselves, the sample size with 77 eTMBC and only 12 mTNBC patients is really small. This reduces the significance of this research paper dramatically.

Also, regrettably, only one type of test has been used making it difficult to transfer these results to patients tested with other PD-L1 assays. Again, this mitigates the importance of the presented results, as especially Atezolizumab is going to be used less due to the Impassion-results (compared to the impressive Keynote-results). Also, when considering that a switch in IHC is a well-known fact, a switch in PD-L1 status is thus not really surprising.

Author Response

Jun 15, 2022

Reviewer #3 (Comments to the Author):

As mentioned in the limitations part by the authors themselves, the sample size with 77 eTNBC and only 12 mTNBC patients is really small. This reduces the significance of this research paper dramatically.

Also, regrettably, only one type of test has been used making it difficult to transfer these results to patients tested with other PD-L1 assays. Again, this mitigates the importance of the presented results, as especially Atezolizumab is going to be used less due to the Impassion-results (compared to the Impressive Keynote-results). Also, when considering that a switch in IHC is a well-known fact, a switch in PD-L1 status is thus not really surprising.

Response) Thank you for your kind and sincere comments. When investigating for this study, unfortunately, the number of patients who could identify PD-L1 status in multiple assays was small. Currently, PD-L1 assays including SP142 or 22C3 are not routinely recommended in a daily practice for patients with early TNBC; thus, it is not easy to collect paired samples from patients with early TNBC. The statistics had to be reflective of this, we could not perform subgroup analyses in various ways as the sample size in a given category will be way too small. Given that it is not an easy task to collect TNBC samples, we can draw meaningful findings in our early TNBC cohort despite small sample size.

In the IMpassion 130 trial, PD-L1 positive was confirmed regardless of time and sample location, which means that the number of patients who can be included in the atezolizumab-based chemotherapy group of the IMpassion 130 trial increases when SP142 PD-L1 assays are performed in multiple samples. Although the number of patients in our study was small, we thought that the necessity of performing vigorous multiple tests was explained through statistical analyses. We expected our study will serve as reference data for the treatment of TNBC patients in today and tomorrow and also we plan to confirm this trend again in a larger cohort in near future.
            As you mentioned, it is true that pembrolizumab has recently drawn more attention over atezolizumab in the TNBC treatment based on KEYNOTE-355&-522 trials and so on. But what we sought to investigate in this study is the discrepancy of PD-L1 status in multiple samples, not the utility of atezolizumab. Although there may be difference in the mechanism of action, our findings with alterations in PD-L1 status in multiple samples from the identical patient may provide a clinical tactic for applying these ICBs for patients with TNBC. understanding the mechanisms of other immunotherapies.
 As mentioned in the limitation section, the fact that only one test method was used and the number of patients was small is a caveat of our study, but we think the strength is that we found the reason for multiple test through statistical analysis. And we also expect that additional analysis via various PD-L1 methods with a larger number of patients will be possible in near future.

We thank the reviewer for your careful reading of our manuscript and your helpful comments and suggestions, which were valuable and helpful for revising and improving our manuscript. We now hope that our revisions meet your requirements for publication.

Sincerely,

Sung Gwe Ahn, MD, PhD

Round 2

Reviewer 1 Report

Please check the following minor comments and homogenous the format throughout the manuscript again before publication.  Good luck.

1) Line 23 and line 174: p should be italic.

2) Line 215 Figure 3: Please italic p in the images.

3) Line 254: Please correct “Figure 34 down”.

4) Line 318: Please delete extra “,”.

5) Figure S1 is not clear; make sure it is high resolution for publication.